# Social prescribing for children and youth: A scoping review protocol

**Caitlin Muhl**[1], **Kate Mulligan**[2]*, **Imaan Bayoumi**[3], **Rachelle Ashcroft**[4], **Amanda Ross-White**[5,6], **Christina Godfrey**[1,6]

1 School of Nursing, Faculty of Health Sciences, Queen's University, Kingston, Ontario, Canada, 2 Dalla Lana School of Public Health, University of Toronto, Toronto, Ontario, Canada, 3 School of Medicine, Faculty of Health Sciences, Queen's University, Kingston, Ontario, Canada, 4 Factor-Inwentash Faculty of Social Work, University of Toronto, Toronto, Ontario, Canada, 5 Bracken Health Sciences Library, Queen's University, Kingston, Ontario, Canada, 6 Queen's Collaboration for Health Care Quality: A JBI Centre of Excellence, Queen's University, Kingston, Ontario, Canada

* kate.mulligan@utoronto.ca

**Data Availability Statement:** No datasets were generated or analysed during the current study. All relevant data from this study will be made available upon study completion.

## Abstract

Social prescribing is suited to all age groups, but it is especially important for children and youth, as it is well understood that this population is particularly vulnerable to the effects of the social determinants of health and health inequities, and that intervening at this stage of life has the greatest impact on health and wellbeing over the life course. While this population has largely been neglected in social prescribing research, policy, and practice, several evaluations of social prescribing for children and youth have emerged in recent years, which calls for a review of the evidence on this topic. Thus, the objective of this scoping review is to map the evidence on the use of social prescribing for children and youth. This review will be conducted in accordance with the JBI methodology for scoping reviews and will be reported in line with the Preferred Reporting Items for Systematic reviews and Meta-Analyses extension for Scoping Reviews (PRISMA-ScR). The search strategy will aim to locate both published and unpublished literature. No language or date restrictions will be placed on the search. The databases to be searched include MEDLINE (Ovid), CINAHL (EBSCO), Embase (Ovid), PsycINFO (Ovid), AMED (Ovid), ASSIA (ProQuest), Sociological Abstracts (ProQuest), Global Health (Ovid), Web of Science (Clarivate), Epistemonikos, JBI EBP Database (Ovid), and Cochrane Library. Sources of gray literature to be searched include Google, Google Scholar, Social Care Online (Social Care Institute for Excellence), SIREN Evidence and Resource Library (Social Interventions Research and Evaluation Network), and websites of social prescribing organizations and networks. Additionally, a request for evidence sources will be sent out to members of the Global Social Prescribing Alliance. Two independent reviewers will perform title and abstract screening, retrieval and assessment of full-text evidence sources, and data extraction. Data analysis will consist of basic descriptive analysis. Results will be presented in tabular and/or diagrammatic format alongside a narrative summary.

**Funding:** The authors received no specific funding for this work.

**Competing interests:** The authors have declared that no competing interests exist.

## Introduction

The COVID-19 pandemic has exacerbated longstanding health inequities in society and unmasked the impact of the social determinants of health (SDOH) [1]. In doing so, this global crisis has shed light on the importance of social prescribing as a way to mitigate the effects of adverse SDOH and health inequities to support the achievement of global goals for health and wellbeing [2–4]. Social prescribing is "a means for trusted individuals in clinical and community settings to identify that a person has non-medical, health-related social needs and to subsequently connect them to non-clinical supports and services within the community by co-producing a social prescription–a non-medical prescription, to improve health and wellbeing and to strengthen community connections" [4 p. 9]. There is a growing body of evidence on social prescribing [5], which suggests that it supports the Quadruple Aim of improved client experience, improved population health, reduced costs, and improved provider experience [2, 6, 7], and that it also advances health equity [3], which is the fifth aim in the newly established Quintuple Aim [8]. There are over 20 countries involved in the social prescribing movement [9], and this number continues to grow.

Social prescribing is especially important for children and youth [10–12], as it is well understood that this population is particularly vulnerable to the effects of the SDOH and health inequities [13], and that intervening at this stage of life has the greatest impact on health and wellbeing over the life course [14]. Despite the clear rationale to target children and youth in social prescribing efforts, this population has largely been neglected in social prescribing research, policy, and practice, with adults receiving most of the attention [10–12]. As a result, evidence reviews on social prescribing have been heavily focused on adults [15–24]. However, several evaluations of social prescribing for children and youth have emerged in recent years, such as the Young People Social Prescribing pilot [25], CHOICES (CHildren and young people's Options In the Community for Enhancing wellbeing through Social prescribing) [26], and INSPYRE (Increasing Social Prescribing Youth Referrals) [27]. Given the growing number of evaluations of social prescribing for children and youth, a review of the evidence on this topic would be most valuable. A preliminary search of PROSPERO, MEDLINE (Ovid), JBI EBP Database (Ovid), Cochrane Library, Google, and Google Scholar was conducted on October 28, 2023. A handful of current and underway evidence reviews on social prescribing for children and youth was found [28–33], albeit with a specific focus on mental health [28–32] and restrictions on language [28, 29, 31, 32], date [33], type of participants [30, 31, 33], type of social prescribing [28, 29, 32, 33], and type of context [33]. Notably, no evidence reviews with an aim to comprehensively map the evidence on social prescribing for children and youth were found. To address this gap in the literature, there is a need for a review of the evidence that is not solely focused on mental health or restricted by language, date, participants, concept, or context. Additionally, with the recent development of internationally accepted conceptual and operational definitions of social prescribing [4], there is an opportunity to use the operational definition to develop inclusion and exclusion criteria that aligns with current understanding of this concept and to build a search strategy that is able to locate relevant evidence sources that are not labelled as social prescribing.

There are many different types of evidence reviews [34]. Scoping reviews are conducted to determine the scope of a body of literature on a particular topic [34–37]. This type of review is particularly useful for reviewing evidence in emerging fields or topics and for addressing broad review questions [35–37], making it the most appropriate method to map the evidence on social prescribing for children and youth. Thus, a scoping review of the evidence will be conducted.

The objective of this scoping review is to map the evidence on the use of social prescribing for children and youth. Recommendations for future research will be made based on identified knowledge gaps.

### Review questions

1. What evidence exists on the use of social prescribing for children and youth?

2. What are the knowledge gaps in the evidence base around the use of social prescribing for children and youth?

### Eligibility criteria

**Participants.** This review will consider evidence sources with participants who are children and youth (≤25 years of age). Evidence sources with participants who are not children and youth (>25 years of age) will be excluded. This age cutoff aligns with other reviews on this topic [29–33]. Participants must be children and youth rather than, for example, families, parents, guardians, or caregivers.

**Concept.** This review will consider evidence sources that explore a concept that meets the following operational definition of social prescribing, even if it is not labelled as social prescribing:

Social prescribing is "a holistic, person-centred, and community-based approach to health and wellbeing that satisfies Condition 1 and either Condition 2 or Conditions 3 and 4:

- Condition 1: Identifier identifies that person has non-medical, health-related social needs (e.g., issues with housing, food, employment, income, social support)

- Condition 2: Identifier connects person to non-clinical supports and services within the community by co-producing a non-medical prescription

- Condition 3: Identifier refers person to connector

- Condition 4: Connector connects person to non-clinical supports and services within the community by co-producing a non-medical prescription" [4 p. 9].

Evidence sources that explore a concept that does not meet this definition will be excluded.

**Context.** This review will consider evidence sources from any context.

**Types of evidence sources.** This review will consider both published and unpublished literature. Evidence sources with quantitative, qualitative, and mixed methods study designs will be considered. In addition to primary research, this review will consider reviews and meta-analyses, theses and dissertations, and reports. Evidence sources without full text, text and opinion papers, and protocols will be excluded.

## Methods

This protocol has been registered on Open Science Framework (osf.io/xhymv) and published in this journal to promote transparency. The creation of this protocol was informed by best practice guidance for scoping review protocols from the JBI Scoping Review Methodology Group [38]. This review will be conducted in accordance with the JBI methodology for scoping reviews [36] and will be reported in line with the Preferred Reporting Items for Systematic reviews and Meta-Analyses extension for Scoping Reviews (PRISMA-ScR) [39]. This review will be published in a peer-reviewed journal. All data relevant to the study will be included in the article or uploaded as S1 File.

### Search strategy

The search strategy will aim to locate both published and unpublished literature. An initial limited search of MEDLINE (Ovid) and CINAHL (EBSCO) was undertaken to identify

**Table 1. Search strategy for MEDLINE (Ovid).**

| Number | Query | Results |
|:---:|:---:|:---:|
| 1 | ((social or non-medical or non-clinical or community) adj (prescri* or referral*)).ti. | 448 |
| 2 | ((social determinant* or social risk* or social need*) and (identif* or screen* or connect* or referral* or address* or navigat*)).ti. | 858 |
| 3 | 1 or 2 | 1302 |
| 4 | exp Child/ | 2167655 |
| 5 | exp Child Health/ | 5183 |
| 6 | exp Child Health Services/ | 25755 |
| 7 | exp Adolescent/ | 2224547 |
| 8 | exp Adolescent Health/ | 1932 |
| 9 | exp Adolescent Health Services/ | 5897 |
| 10 | exp Young Adult/ | 1016158 |
| 11 | exp Pediatrics/ | 63157 |
| 12 | (child* or adolescen* or youth or young people* or young person* or young adult* or teen* or pediatric* or paediatric*).mp. | 4571010 |
| 13 | 4 or 5 or 6 or 7 or 8 or 9 or 10 or 11 or 12 | 4575056 |
| 14 | 3 and 13 | 299 |

Search conducted on October 31, 2023

evidence sources on the topic. The text words contained in the titles and abstracts of relevant evidence sources, and the index terms used to describe the evidence sources, were used to develop a full search strategy for MEDLINE (Ovid) in Table 1. The search strategy was developed in consultation with an academic health sciences librarian. The research team will adapt the search strategy, including all identified keywords and index terms, for each included database and source of grey literature. This will be done in consultation with the academic health sciences librarian. The reference lists of all included evidence sources will be screened for additional evidence sources. No language or date restrictions will be placed on the search. In the event that translation becomes necessary, the review team will use DeepL Translator (DeepL SE, Cologne, Germany).

The databases to be searched include MEDLINE (Ovid), CINAHL (EBSCO), Embase (Ovid), PsycINFO (Ovid), AMED (Ovid), ASSIA (ProQuest), Sociological Abstracts (ProQuest), Global Health (Ovid), Web of Science (Clarivate), Epistemonikos, JBI EBP Database (Ovid), and Cochrane Library. Sources of gray literature to be searched include Google, Google Scholar, Social Care Online (Social Care Institute for Excellence), SIREN Evidence and Resource Library (Social Interventions Research and Evaluation Network), and websites of social prescribing organizations and networks, including the Social Prescribing Network, the Social Prescribing Youth Network, the Global Social Prescribing Alliance, the National Academy for Social Prescribing, and the Canadian Institute for Social Prescribing. Additionally, a request for evidence sources will be sent out to members of the Global Social Prescribing Alliance.

## Evidence source selection

Following the search, all identified evidence sources will be collated and imported into Covidence (Veritas Health Innovation, Melbourne, Australia) and duplicates removed. Following a pilot test, titles and abstracts will then be screened by two independent reviewers for assessment against the inclusion criteria for the review. Potentially relevant evidence sources will be retrieved in full, imported into Covidence, and assessed in detail against the inclusion criteria

**Table 2. Draft data extraction tool.**

| Citation Details | Objective | Participants | Concept | Context | Methods | Key Findings |
|---|---|---|---|---|---|---|
|  |  |  |  |  |  |  |

by two independent reviewers. Reasons for exclusion of evidence sources at full text that do not meet the inclusion criteria will be recorded and reported in the scoping review. Any disagreements that arise between the reviewers at each stage of the selection process will be resolved through discussion or with a third reviewer. The results of the search and the evidence source inclusion process will be reported in full in the final scoping review and presented in a PRISMA-ScR flow diagram [39].

## Data extraction

Data will be extracted from evidence sources included in the scoping review by two independent reviewers using a data extraction tool developed by the research team. The data extracted will include specific details about the objective, participants, concept, context, methods, and key findings relevant to the review questions. A draft data extraction tool is provided in Table 2. This tool will be modified and revised as necessary during the process of extracting data from each included evidence source. Modifications will be detailed in the scoping review. Any disagreements that arise between the reviewers will be resolved through discussion or with a third reviewer. Authors of evidence sources will be contacted to request missing or additional data, where required.

## Data analysis and presentation

Data analysis will consist of basic descriptive analysis. Results will be presented in tabular and/or diagrammatic format in a manner that aligns with the review objective. A narrative summary will accompany the tabulated and/or charted results and will describe how the results relate to the review objective and questions.

## Supporting information

**S1 File. Scoping review protocol checklist.**
(PDF)

## Author Contributions

**Conceptualization:** Caitlin Muhl.

**Writing – original draft:** Caitlin Muhl.

**Writing – review & editing:** Caitlin Muhl, Kate Mulligan, Imaan Bayoumi, Rachelle Ashcroft, Amanda Ross-White, Christina Godfrey.

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
