## [Decision Letter · Decision Letter 0]

27 Oct 2023

PONE-D-23-11764

Social prescribing for children and youth: A scoping review protocol

PLOS ONE

Dear Dr. Muhl,

Thank you for submitting your manuscript to PLOS ONE. After careful consideration, we feel that it has merit but does not fully meet PLOS ONE’s publication criteria as it currently stands. Therefore, we invite you to submit a revised version of the manuscript that addresses the points raised during the review process.

We look forward to receiving your revised manuscript.

Kind regards,

Olujide Arije

Academic Editor

PLOS ONE

Journal Requirements:

3. We note that this manuscript is a systematic review or meta-analysis; our author guidelines therefore require that you use PRISMA guidance to help improve reporting quality of this type of study. Please upload copies of the completed PRISMA checklist as Supporting Information with a file name “PRISMA checklist”.

Additional Editor Comments:

The reviewers generally concur on the valid rationale behind the protocol's topic. One of the reviewers indicate that additional justification of the publication of the protocol is required. However, no modifications are required to address this concern.

The editorial staff has brought to my attention that there is a related Study Protocol currently under consideration at PLOS ONE titled "Social Prescribing and Students: A Scoping Review Protocol," which delves into the examination of evidence regarding social prescribing and students. It is essential to emphasize that the journal strongly discourages the unnecessary fragmentation of related work into separate manuscripts. PLOS ONE's policy mandates that each submission be a self-contained unit, without reliance on work that has not yet been accepted for publication. If authors choose to submit related manuscripts to PLOS ONE, they are strongly encouraged to assess the potential for consolidation into a single comprehensive manuscript. Alternatively, authors should provide compelling reasons for separate publication. In view of this, a major revision decision has been given to your manuscript at this time.

Reviewers' comments:

Reviewer's Responses to Questions

**Comments to the Author**

1. Does the manuscript provide a valid rationale for the proposed study, with clearly identified and justified research questions?

Reviewer #1: Yes

Reviewer #2: Partly

Reviewer #3: Yes

2. Is the protocol technically sound and planned in a manner that will lead to a meaningful outcome and allow testing the stated hypotheses?

Reviewer #1: Yes

Reviewer #2: No

Reviewer #3: Yes

3. Is the methodology feasible and described in sufficient detail to allow the work to be replicable?

Reviewer #1: Yes

Reviewer #2: No

Reviewer #3: Yes

4. Have the authors described where all data underlying the findings will be made available when the study is complete?

Reviewer #1: Yes

Reviewer #2: No

Reviewer #3: No

5. Is the manuscript presented in an intelligible fashion and written in standard English?

Reviewer #1: Yes

Reviewer #2: Yes

Reviewer #3: Yes

6. Review Comments to the Author

You may also provide optional suggestions and comments to authors that they might find helpful in planning their study.

Reviewer #1: Social prescribing has been implemented as a novel intervention to address non-clinical social needs with promising results.Children and adolescents do have non clinical social needs and it is important that literature on this intervention in this population is explored.

A scoping review is appropriate in this case. The objectives and methods have been well explained

Reviewer #2: The protocol does have a valid rationale, however, it is not clear why there is a need for the authors to have their protocol published before they start with their actual reviews. I do not see this as a topical issue that can be at risk if not published from the protocol.

The methodology is not clearly outlined and described, and this makes it difficult to fully understand the protocol.

Reviewer #3: This is a clear and well written protocol for this scoping review. The authors rightly point out that less attention has been given to this age group and therefore this review is warranted. Furthermore, it will be valuable to apply the operational constructs of their recently published definition to the included studies.

There was a brief description of how data will be presented, but I couldn't see where data underlying the findings will be made available once the study is complete.

I have no further comments and wish the authors good luck with their work.

7. PLOS authors have the option to publish the peer review history of their article (what does this mean?). If published, this will include your full peer review and any attached files.

Reviewer #1: No

Reviewer #2: No

Reviewer #3: **Yes: **Marie Polley

---

## [Author Response · Author response to Decision Letter 0]

30 Nov 2023

Dear Editor,

We note that this manuscript is a scoping review protocol. Therefore, we did not include a PRISMA checklist because that checklist is specifically for systematic review and meta-analysis protocols. Instead, we included a checklist that was developed by JBI for scoping review protocols. This checklist has been included as Supporting Information with a file name “S1_File”.

Kind regards,

Dr. Kate Mulligan

---

## [Decision Letter · Decision Letter 1]

9 Jan 2024

Social prescribing for children and youth: A scoping review protocol

PONE-D-23-11764R1

Dear Dr. Mulligan,

We’re pleased to inform you that your manuscript has been judged scientifically suitable for publication and will be formally accepted for publication once it meets all outstanding technical requirements.

Kind regards,

Dr. Olujide Arije, PhD

Academic Editor

PLOS ONE

Additional Editor Comments (optional):

Reviewers' comments:

Reviewer's Responses to Questions

**Comments to the Author**

1. Does the manuscript provide a valid rationale for the proposed study, with clearly identified and justified research questions?

Reviewer #3: Yes

2. Is the protocol technically sound and planned in a manner that will lead to a meaningful outcome and allow testing the stated hypotheses?

Reviewer #3: Yes

3. Is the methodology feasible and described in sufficient detail to allow the work to be replicable?

Reviewer #3: Yes

4. Have the authors described where all data underlying the findings will be made available when the study is complete?

Reviewer #3: Yes

5. Is the manuscript presented in an intelligible fashion and written in standard English?

Reviewer #3: Yes

6. Review Comments to the Author

You may also provide optional suggestions and comments to authors that they might find helpful in planning their study.

Reviewer #3: I find the author responses acceptable to the comments posed to them. As an expert in this field I would like to highlight the benefit of publishing the protocol as there are many other active research groups and there are few opportunities to know what each other are doing. Publishing this protocol will is useful to ensure no one else starts this work and duplicates time and funding.

7. PLOS authors have the option to publish the peer review history of their article (what does this mean?). If published, this will include your full peer review and any attached files.

Reviewer #3: **Yes: **Dr Marie Polley

---

## [Editor Report · Acceptance letter]

27 Feb 2024

PONE-D-23-11764R1 

PLOS ONE

Dear Dr. Mulligan, 

I'm pleased to inform you that your manuscript has been deemed suitable for publication in PLOS ONE. Congratulations! Your manuscript is now being handed over to our production team.

Kind regards, 

on behalf of

Dr. Olujide Arije 

Academic Editor

PLOS ONE